# Personalised Medicine Using N-of-1 Trials: Overcoming Barriers to Delivery

**DOI:** 10.3390/healthcare7040134

**Published:** 2019-11-05

**Authors:** Iain Chalmers, Liam Smeeth, Ben Goldacre

**Affiliations:** 1Department of Primary Care, University of Oxford, Oxford OX2 6GG, UK; ben.goldacre@phc.ox.ac.uk; 2London School of Hygiene and Tropical Medicine, London WC1E 7HT, UK; liam.smeeth@lshtm.ac.uk

**Keywords:** N-of-1, trials, barriers, implementation, methods, personalised healthcare, hyper-regulation of clinical research

## Abstract

In this paper we discuss the value of N-of-1 trials in personalising health care. We describe the challenges faced in implementing N-of-1 trials in the United Kingdom’s National Health Service and suggest how making greater use of these personalised trials might be facilitated.

## 1. Introduction

There is currently great enthusiasm to improve patient care through ‘personalised medicine’, ‘precision medicine’, or ‘stratified medicine’—targeting therapies to individuals on the basis of genetic or other characteristics. Personalised medicine is not a new notion. There have been arguments for two centuries about the relevance to individual patients of statistical information derived from larger groups [1]. Historically, the notion of tailored treatment was based on clinicians’ claims that they had an intuitive professional understanding of their patients’ needs. Today such claims are sometimes being made by reference to information about a patient’s genetic characteristics.

Individual patients may indeed react differently to a treatment which has been shown to have useful effects when studied in groups; and some aspect of their genetic makeup may indeed sometimes account for these idiosyncrasies, alongside co-prescribing, co-morbidity, preferences, and more. In practice, when a clinician finds that a patient is not responding as predicted to an analgesic for example, she or he will suggest switching (crossing over) to an alternative analgesic. Serious scientific development around these informal ‘crossover trials’ began only a few decades ago; but well designed and executed randomised N-of-1 trials quite properly now occupy the apex of the hierarchy of research designs for informing treatment choices [2,3]. Taking personalisation a step further, outcomes in N-of-1 trials can be chosen by patients. An example of a patient-generated outcome measure is the MYMOP (Measure Yourself Medical Outcome Profile), used in a number of N-of-1 trials [4].

## 2. N-of-1 Trial Services

In their review of the history of N-of-1 randomised trials, Mirza and her colleagues reviewed services to enable clinicians and patients to run N-of-1 trials. The first of these was a referral service at McMaster University in Canada [5]. Within two years, 57 N-of-1 trials had provided definite therapeutic answers in 88% of the patients studied, and these results prompted 39% of physicians to change their previous treatment plan. An N-of-1 trial service at the University of Washington completed 34 trials. Within two years, the service had demonstrated that such trials could provide useful treatment guidance and improve patient satisfaction [6].

In 1999, the University of Queensland in Australia created the first national N-of-1 trial research service [7]. Physicians could refer their patients to the centrally located service, which managed all major components of trial management: randomization, preparing tablets, sending all materials to patients, following-up, and relaying results to clinicians. Post-trial management decisions were consistent with trial results at 12 months in approximately 70% of ADHD trials [8], 45% of osteoarthritis trials [9], and 32% of neuropathic pain trials [10].

Similar support was instituted by the Complementary and Alternative Research and Education (CARE) program at the University of Alberta when it established the first academic paediatric integrative medicine programme in Canada [11]. This service has assessed natural health products (e.g., melatonin, probiotics, and micronutrients) and acupuncture for conditions including ADHD, eczema, sleep disturbances, chemo-induced nausea and vomiting, irritable bowel syndrome, and autism.

## 3. Establishing a Framework for Running N-of-1 Trials of Statins to Identify or Rule Out Side Effects

In June 2014, in a response to the debate about possible adverse effects of statins, one of us (I.C.) expressed a wish to participate in an N-of-1, placebo-controlled, randomized, crossover trial to assess whether his muscle symptoms were associated with taking atorvastatin [12]. The wish was reiterated a few months later in an article requested by *The Guardian* [13].

In response to people facing uncertainties like I.C.’s, two of us (L.S. and B.G.) worked with others to obtain support for 200 such N-of-1 trials, using outcome measures that reflect symptoms known to be of concern to patients [8]. Participants in the trials were people taking statins who were experiencing unwelcome symptoms that may be caused by statins, and therefore thinking about stopping taking their medications. They had been randomised to a series of two-month blocks of either atorvastatin 20mg or to placebo. Outcomes were adverse symptoms, self-reported using a smart phone app, online or with paper forms, depending on patient preference. The study is registered at controlled-trials.com ISRCTN30952488, August 2016, and at clinicaltrials.gov with the reference NCT02781064 [14]

It was planned to do this trial very inexpensively, as shown to be possible using previous pragmatic randomised controlled trials based within general practices contributing to the Clinical Practice Research Datalink database [15]. However, there were formidable barriers and bureaucratic hurdles that led to substantial delays and expense. As participants and clinicians were blinded to treatment allocation, and the study included time periods of treatment with a placebo medication, this immediately meant the study was classified as a Clinical Trial of an Investigational Medicinal Product (CTIMP). This was in spite of the fact that the trial was using atorvastatin within its licensed indications, and everyone in the trial had previously been prescribed long-term statin therapy. CTIMPs are subject to substantial legislation and scrutiny as to how they are approved, conducted and monitored. While such a high level of regulation is appropriate for new drugs of unknown toxicity, statins are one of the most widely used medications worldwide with a huge wealth of evidence on their effects. One consequence of this regulation was the insistence by the UK regulator - the Medicines and Healthcare products Regulatory Agency—that all patients should have a blood test prior to joining the trial, with no clear clinical justification. The addition of a “research only” blood test that was above and beyond usual clinical care complicated the study, added to costs, and made recruitment more difficult.

Despite valiant efforts by many people, and some clear progress with the creation in the UK of the Integrated Research Application System, obtaining approval to run the study has been a formidable task. In order to be able to recruit participants, general practitioners need formal approval as “sites”. Despite the best efforts of I.C.’s general practitioner, his hope of clarifying whether his symptoms are statin-related have been dashed because the arrangements require complete GP group practices to sign up to the research, and the practice with which he is registered simply does not currently have the capacity to support the bureaucratic demands associated with embedding research within practice.

The UK government has stated repeatedly that it wishes to encourage research to become an integral component of routine practice. Because of hyper-regulation and the want of a supply of placebos, it is infinitely easier for I.C. and his GP to do their own informal, scientifically sub-optimal, crossover trial without placebo controls, with the consequent generation of evidence of questionable validity.

## 4. Promoting Personalised Care by Establishing N-of-1 Trials Services

Calls for the promotion of personalised medicine cannot be taken seriously while it remains so difficult to incorporate scientifically trustworthy N-of-1 trials to address questions that are of importance to help inform the treatment choices of individual patients. In our view the barriers to such work are cultural, practical, and regulatory, rather than scientific, ethical or technical. However, if N-of-1 trials are to be more widely used to inform clinical practice, it is clear that barriers to their implementation will need to be overcome.

We suggest that the following practical components are required, at a national level, to facilitate true ‘personalised medicine’ through routine use of N-of-1 trials:A research pharmacy to encapsulate, number and label two lots of medication: some placebo and some the drug.A simple randomisation system for the crossover, with appropriate time periods tailored to the clinical question, and to number the capsule packs.A flexible method to collect repeated self-reports of possible beneficial and unwanted effects (such as pain relief or gastrointestinal symptoms).Some ‘back-end’ online software to process, analyse and present the data to clinicians and patients.

The main downside of this plan is that, with too few crossovers or too few measures of symptoms, there will be a fairly high chance of incorrectly asserting an untrue relationship between drugs and symptoms. A key upside is that, with sufficiently large numbers of individuals’ trials on the same clinical question, one could retrospectively aggregate all the data to yield statistically more robust estimates of treatment effects.

A simple practical approach to delivering N-of-1 trials as a clinical investigation has the potential to substantially improve the quality and cost-effectiveness of care and to deliver personalised medicine. However, considerable effort and some investment is required to overcome existing barriers to the use of such trials in everyday clinical practice.

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
