# Peer review of "Personalised Medicine Using N-of-1 Trials: Overcoming Barriers to Delivery"

_healthcare, 2019, doi:10.3390/healthcare7040134_

Round 1

Reviewer 1 Report

The comparison with the NCI-MATCH and the SHIVA trial is somewhat overstating the issue. In these trials (heavily pretreated) cancer patients are receiving molecularly targeted agents outside of the prescription. This is entirely different from the situation discussed by the authors for statins, i.e., the comparison is unjustified. Nevertheless, the authors do point out an important matter: how to perform a straightforward trial like the one suggested, without being caught in the regulatory network. Indeed, as written by the authors the barriers to such work are cultural, practical, and regulatory, rather than scientific or technical. The practical adjustments suggested may take away some of the practical hurdles, but others remain: classification as a CTIMP, which introduces administrative and practical hurdles, and the need for a complete GP group to sign up for the research, is still not addressed. In view of the former it would be advantageous to pay somewhat more attention to these issues and discuss possibilities to advance this field also in this area.

Author Response

Thanks you for the useful comments.

We have now modified the content about personalised medicine in line with the referee’s comments.

We have also more directly addressed the main barriers to n of 1 trials as suggested.

Reviewer 2 Report

Healthcare-599041 – Review Personalised medicine using n-of-1 trials: overcoming barriers to delivery

I very much welcome this short article which covers some of the pressing issues about n-of-1 trials. This kind of trial has enormous potential: to identify treatments which do actually help individuals; to thereby avoid waste; to support shared decision making; to address research questions identified by patients (as in this article); to include patient preferences in research; and more. The article itself is original in bringing together the perspective of a willing potential participant in such a trial, with the experience of running a well-funded study from a prestigious research organisation. When I heard about the award of this grant from the Health Technology Assessment programme in the UK, I was heartened to conclude that n-of-1 trials had reached a greater level of recognition. The significance of the article is in the fact that this trial, despite being funded after rigorous peer review, ran into bureaucratic obstacles which hindered the research for no good reason. This may well contribute to the debate about research into n-of-1 trials, and the possibility of generating high quality evidence from this method.

While I very much agree with the authors’ arguments, there seems to be a disconnect between the problems described in the article and the solutions proposed. The problems described are about conducting research about n-of-1 trials, while the solutions proposed are relevant for anyone who wants to conduct a rigorous n-of-1 trial in clinical practice. So I would like to see some proposed solutions to the problems of doing research in this field, such as the regulations surrounding CTIMPs. I would also like to see some mention of the practical problems which the proposed solutions aim to address.

It is my understanding, although the authors will know much better than I do, that in the past Australian researchers did manage to set up a research pharmacy to produce and package active treatments and placebos in support of n-of-1 trials. If I am right, it would be good to include a reference to this.

I would also like to see at least a mention of the possibility of patient-chosen outcome measures (I am not talking about patient-reported outcome measures which are something different, because they are chosen and designed by academics and clinicians). If the rationale behind n-of-1 trials is to find a treatment which works for an individual patient, it is important to measure the criterion (or criteria) by which that individual judges the treatment to have ‘worked’. This could include generic questions about interference with everyday life for example. There are several examples of patient-generated outcome measures which might be suitable. One of them, MYMOP (Measure Yourself Medical Outcome Profile), has been used in a number of n-of-1 trials already (Paterson 2004).

To declare an interest: I have experienced similar bureaucratic obstacles when conducting a trial of a non-invasive, non-drug intervention, which severely hampered recruitment. I currently have a manuscript under review, but I think these kinds of obstacles to trials of treatments or interventions which have already been licensed, which are non-invasive, or which are embedded in everyday clinical practice, deserve urgent attention.

The paper is well written and would be of great interest to the journal’s readers.

Reference

C Paterson (2004) Seeking the patient’s perspective: a qualitative assessment of EuroQoL, COOP-WONCA Charts and MYMOP2. Qual Life Res; 13:871-81.

Author Response

Thank you for the useful comments.

We have now more directly addressed the main barriers to n of 1 trials as suggested.

We have included the issue of patient generated outcome measures.